# Semantic Segmentation of Remote Sensing Data Based on Channel Attention and Feature Information Entropy

**DOI:** 10.3390/s24041324

**Published:** 2024-02-19

**Authors:** Sining Duan, Jingyi Zhao, Xinyi Huang, Shuhe Zhao

**Affiliations:** 1Jiangsu Provincial Key Laboratory of Geographic Information Science and Technology, Key Laboratory for Land Satellite Remote Sensing Applications of Ministry of Natural Resources, School of Geography and Ocean Science, Nanjing University, Nanjing 210023, China; s70170569@163.com (S.D.); mg21270103@smail.nju.edu.cn (X.H.); 2Jiangsu Center for Collaborative Innovation in Geographical Information Resource Development and Application, Nanjing 210023, China; 3School of Information Science and Engineering, China University of Petroleum (Beijing), Beijing 102249, China; 2020011631@student.cup.edu.cn

**Keywords:** channel attention mechanism, land use classification, semantic segmentation

## Abstract

The common channel attention mechanism maps feature statistics to feature weights. However, the effectiveness of this mechanism may not be assured in remotely sensing images due to statistical differences across multiple bands. This paper proposes a novel channel attention mechanism based on feature information called the feature information entropy attention mechanism (FEM). The FEM constructs a relationship between features based on feature information entropy and then maps this relationship to their importance. The Vaihingen dataset and OpenEarthMap dataset are selected for experiments. The proposed method was compared with the squeeze-and-excitation mechanism (SEM), the convolutional block attention mechanism (CBAM), and the frequency channel attention mechanism (FCA). Compared with these three channel attention mechanisms, the mIoU of the FEM in the Vaihingen dataset is improved by 0.90%, 1.10%, and 0.40%, and in the OpenEarthMap dataset, it is improved by 2.30%, 2.20%, and 2.10%, respectively. The proposed channel attention mechanism in this paper shows better performance in remote sensing land use classification.

## 1. Introduction

The spatial resolution of remote sensing images has improved continuously in recent years [1]. The details of objects in images are more obvious, but this also causes more serious intra-class spectral differences and inter-class spectral similarity in image data. This reduces separability in the spectral domain, causing a great deal of trouble for land use classification [2]. Traditional statistical classification methods, such as the maximum likelihood method, and machine learning algorithms, such as support vector machine and random forest, struggle to classify such images effectively [3].

Semantic segmentation is a method that extracts features from images based on their spatial–spectral and contextual information. It then uses these features to classify each pixel in the image. Therefore, semantic segmentation is applied to process high-resolution remote sensing images, which are increasingly rich in spatial information. Many excellent semantic segmentation algorithms have emerged, such as the FCN, Unet, Segnet, and Deeplab series [4,5,6,7]. These methods have achieved good results in land use classification compared to traditional machine learning algorithms.

Semantic segmentation usually includes two stages: encoder and decoder. Many scholars have made efforts to improve the performance of the encoder. Hu et al. designed a channel attention mechanism called the squeeze-and-excitation module (SEM) [8]. This mechanism explicitly models the interdependence between features according to the global average value of features. The relationships are then used to nonlinearly scale features. It enhances important features and suppresses invalid features to improve the effectiveness of the encoder. Woo et al. proposed convolutional block attention module (CBAM) [9]. In this mechanism, the global maximum value and the global average value of features are chosen as the basis for feature enhancement. It produces more refined channel attention and applies it to the semantic segmentation field. Qin followed the idea of enriching feature statistics and conducted a deep study on the feature statistics selection of channel attention mechanisms. He regarded the process of obtaining feature statistics as a compression process using frequency analysis and proposed a frequency channel attention (FCA) mechanism [10]. This attention mechanism achieved a 1.8%-higher Top-1 accuracy on the ImageNet dataset compared to the SEM. Fu et al. changed the design idea behind channel attention mechanisms and proposed the dual attention mechanism (DAM) [11]. Each high-level feature obtained via the convolutional network is regarded as a response to different objects in the original image. Taking these into account, the mechanism then enhances only the high-level features. Benefiting from the small feature size of the highest-level features, feature weights are directly determined by the highest-level features themselves.

As different channel attention mechanisms are proposed, they are also being applied to remote sensing land use classification. Panboonyuen et al. introduced the SEM into remote sensing semantic segmentation, enhancing its accuracy. Building upon this advancement, they further integrated additional mechanisms with the encoder, also aiming to boost its accuracy [12]. Abdollahi et al. added the SEM to the Unet network, and it provided certain assistance in the decoder [13]. Lan et al. adopted Deeplab architecture and added the SEM in atrous spatial pyramid pooling (ASPP) to deal with high-level features [14]. Yang et al. introduced the CBAM into the Unet network and used this mechanism not only in the encoder but also in the decoder. They also used other mechanisms, such as ASPP, to assist feature extraction [15]. Zhang et al. used the CBAM but only operated on the high-level features extracted by Resnet [16]. Liu et al. used the DAM to enhance feature extraction [17]. Hou et al. proposed the mean weighted module (MWM), aiming to efficiently establish dependencies between channels while preserving spatial feature information and overall feature dimensionality [18]. Wang et al. proposed the vector pooling attention (VPA) module for enhancing the channel attention mechanism to better retain spatial information and establish long-range dependencies [19].

The popularity of channel attention mechanisms in the field of remote sensing semantic segmentation is evident. However, channel attention mechanisms are usually directly applied to the remote sensing field. There is a lack of improvement for remote sensing images. Remote sensing images are different from natural images in that natural images only contain information in visible wavelengths, while remote sensing images often contain near-infrared (NIR) wavelengths. Nevertheless, the near-infrared band and the visible light band each contain different information and exhibit a strong non-correlation. Common channel attention mechanisms usually map feature statistics such as average and maximum to feature weights. High reflectance in the NIR for land use classes such as vegetated, bare land, and artificial surfaces can lead to higher average values than in the visible band. With the NIR band having the highest average value, it might appear that the NIR band warrants additional focus. This discrepancy could potentially disrupt the evaluation of weights by the channel attention mechanism. Moreover, observing the maximum values of each band potentially leads to disproportionate emphasis on one of these classes at the expense of other classes.

Aiming at the above prominent problems. This paper proposed a channel attention mechanism based on feature information content named the feature information entropy attention mechanism (FEM). The information content in each feature is utilized by the FEM to assess the importance of features. By processing the varying importance of features, the FEM combines the different information between visible light and near infrared. So, the function of enhanced feature extraction is finally achieved. At the end of the article, the effectiveness of the channel attention mechanism and the correctness of the determination of feature weights are also analyzed in terms of their interpretability. Feature importance evaluation in remote sensing images can be effectively achieved through the FEM. Furthermore, visualizations of feature weights and feature downscaling can verify the accuracy of feature weights and facilitate a better design of the channel attention mechanism.

## 2. Methods

### 2.1. Squeeze-and-Excitation Module

The squeeze-and-excitation module (SEM) was initially applied to scene recognition [8]. The good performance of the SEM soon led to its application in many fields. The SEM is what we call the channel attention mechanism. The mechanism determines the importance of features based on their average value. The details of the calculation are as follows: First, features are compressed, which is also known as global information aggregation. This means that the averages of the features are calculated and stored as a vector of statistics with a size of 1 × 1 × c. Next, using the information aggregated during compression, the vector of statistics is input to a multilayer perceptron that captures the feature relationship. Finally, the softmax activation function is applied to the relationship to obtain the feature weights. Then, feature enhancement is completed. Figure 1 shows the SEM.

### 2.2. Convolutional Block Attention Mechanism

The CBAM contains both spatial and channel attention mechanisms, but in this paper, only the channel attention of the CBAM is used for comparative study. The mechanism goes one step further than the SEM by making a feature importance judgement based on the maximum and average values of the features. The details are demonstrated as follows: First, we calculate the global average and maximum for each feature and then obtain a 1 × 1 × C sized statistic vector. To take advantage of the information gathered during the compression operation, the statistic vector is then fed into a shared multilayer perceptron to capture the correlations among features. Finally, the softmax activation function is used to obtain the weights of each feature and to complete the enhancement of the features. The CBAM module is shown in Figure 2.

### 2.3. Frequency Channel Attention Mechanism

The FCA continues the upgrade of the SEM by the CBAM and continues to enrich the feature statistics. The frequency information of the features obtained through discrete cosine transform (DCT) is used to judge the feature importance. The details are demonstrated as follows: First, the features are divided into n blocks along the channel, and the DCT transform is applied to each block to obtain a frequency vector of size 1 × 1 × C/n. The obtained frequency vector is then fed into a multilayer perceptron to capture the correlations among features. Finally, the weights of each feature are obtained using the softmax activation function and the enhancement of the features is conducted via the dot product of the weights and the features. The FCA module is shown in Figure 3.

### 2.4. The Proposed Channel Attention Mechanism

This paper proposes a channel attention mechanism named the feature information entropy attention mechanism. The mechanism represents an improvement on feature information compression and feature weights calculation. In the process of compressing feature information, i.e., the calculation of feature statistics, both the average value and the maximum value of features lack the physical meaning behind them. Furthermore, these values do not adequately represent the entirety of the feature. In order to select a suitable feature statistic, we start from interpretability. The importance of the feature can be measured using the information content of the feature, a metric with clear physical meaning. Thinking in terms of information content, we usually think that visible light bands have high correlation with each other and have redundant information. The NIR bands have different feature information from the visible light bands, so they should be focused on in the feature extraction stage to improve the effect of feature extraction. Shannon entropy is a widely recognized method used to measure the information content:(1)$$H\left(x\right)=-\int_{x}P\left(x_{i}\right)\log p\left(x_{\dot{i}}\right)dx_{i},$$
where $P\left(x_{i}\right)$ is the probability of $x_{i}$ and $x_{i}$ is each pixel value in the feature.

However, it is challenging to calculate the probability distribution of each feature. The article “Opening the black box of deep neural networks via information” assumed that features all belong to the Gaussian distribution [20]. Formula (2) shows how to calculate the information entropy of Gaussian distributions. Through Formula (2) it can be found that the information content of a feature can be represented by its variance. Therefore, the variances of the features are used as a basis for measuring the importance of features.
(2)$$H\left(N\left(\mu ,\sigma^{2}\right)\right)=\frac{1}{2}{log}_{2}2\pi e\sigma^{2}$$

The SEM proposed two guidelines to accurately determine the importance of features based on their statistics: first, it must be flexible (in particular, it must be able to learn nonlinear interactions between features); second, it must excavate non-mutually exclusive relationships, ensuring that multiple features are allowed to be activated, rather than only one feature being activated [8]. The multilayer perceptron (MLP) was chosen by the SEM for this task. MLP builds indirect relationships between two features. This indirect relationship is not as convenient and accurate as building direct relationships between features. Based on the above guidelines, compressed dot-product attention was used to mine the relationships between features. The compressed dot product attention mechanism can relate the features statistics to calculate their nonlinear correlation while guaranteeing that multiple features are enhanced [21]. In detail, the FEM is represented as a function of three vectors query, keys, and value.
(3)$$Attention\left(Q,K,V\right)=Softmax\left(\frac{QK^{T}}{\sqrt{d_{K}}}\right)V$$

The vectors query, keys, and value are mapped from the feature information entropy P ∈ b × c × 1 mentioned above. Then, the transpose of the query vector and keys vector is multiplied to obtain the feature relationship. Finally, the relationship vector is multiplied by the value vector and then activated by the Sigmoid function to obtain the feature weights. The processing is shown in Figure 4.

### 2.5. Backbone Network

The backbone network in this paper is Unet, on which the performance comparisons of different kinds of attention mechanisms are based. The channel attention mechanism is selected for use by checking the mean and variance of the feature weights before each downsampling. In this study, the threshold of variance is set to 0.00025 and the threshold of mean is 0.667. The fact that both the mean and variance of the weights are satisfied means that the channel attention mechanism considers all the features to be equally important. There is no need for the attention mechanism to perform additional enhancement of the features. The reason why we “choose to use the channel attention mechanism” is that in a series of experiments (shown later in the article), the channel attention mechanism does not necessarily work throughout the entire feature extraction process; sometimes, it takes only one feature weights determination to affect the whole feature extraction process and achieve good results. In this case, the demand can be satisfied without a lot of channel attention enhancement. The overall architecture proposed in this paper is shown in Figure 5.

### 2.6. Accuracy Metrics

The overall accuracy, *IoU*, and *mIoU* are selected as accuracy metrics, and the formulae of these accuracy metrics are shown below:(4)$$OA=\frac{\sum_{k=1}^{N}TP_{k}+TN_{k}}{\sum_{k=1}^{N}TP_{k}+FP_{k}+TN_{k}+FN_{k}},$$
(5)$${IoU}_{k}=\frac{TP_{k}}{TP_{k}+FP_{k}+FN_{k}},$$
(6)$$mIoU=\frac{\sum_{k=1}^{N}TP_{k}}{\sum_{k=1}^{N}TP_{k}+FP_{k}+FN_{k}},$$
where *TP_k_*, *FP_k_*, *TN_k_*, and *FN_k_* denote the number of true positive, false positive, true negative, and false negative pixels for object indexed as class *k*, respectively.

## 3. Experiments and Results

### 3.1. Datasets

There has been a lot of work on land use classification using semantic segmentation, and there are many relevant datasets. According to the study by Nayak et al. [22], experiments are conducted on two datasets, the Vaihingen dataset and the OpenEarthMap dataset. The two datasets have different resolutions, and the experiments are carried out on datasets with large resolution differences in order to better evaluate the performance of different channel attention mechanisms.

The Vaihingen dataset has a resolution of 9 cm, including R, G, B, and NIR, four bands and DSM data with the same resolution of 9 cm. The dataset contains a total of 33 images, of which 16 comprise the training set and 17 comprise the test set. It is divided into six categories: “background, impervious surface, building, tree, grassland, car” [23]. NIR, R, and G band data were selected for experiments. A piece of the experimental data is shown in Figure 6.

The OpenEarthMap dataset has a resolution of 0.25–0.5 m, including RGB three bands. A piece of the experiment data is shown in Figure 7. The whole dataset consists of 5000 aerial images, covering 44 regions of 97 countries in five continents [24]. However, due to its large data volume, this experiment only used data from the German region. The selected datasets include 72 images in the training set and 12 images in the test set. The selected data include seven categories: developed land, tree, road, agricultural land, grassland, water body, and building. A piece of the experimental data is shown in Figure 7.

### 3.2. Experimental Setting

First, the experiments compared the performance of the SEM, DAM, FEM, and the baseline architecture Unet under the original mode. Second, the experiments compared the performance of different modes of the channel attention mechanism.

Then, the experiments tested whether different channel attention mechanisms can correctly map feature statistics to feature weights, i.e., the effectiveness of the channel attention mechanisms. This part of the experiment is divided into two phases: comparing feature weights and analyzing the interpretability of the attention mechanisms, respectively.

#### 3.2.1. Experimental Part 1

We consider setting a tensor of size 1 × C × 1 × 1 (C is the number of channels) with all values being 1 to replace the feature statistics. This tensor serves as the foundation for assessing feature importance via the channel attention mechanism. Figure 8 and Figure 9 demonstrate the feature entropy attention (One-FEM) and squeeze excitation mechanism (One-SEM) that replaced the feature statistics, respectively. By comparing the improvement in accuracy between the normal channel attention mechanism and the attention mechanism with replaced statistics, it can be verified that the enhancement of the performance of the channel attention mechanism is due to the enhancement of its important features. If the feature weights are mapped from a tensor with all values being 1, but the mechanism still produces a similar performance to the original channel attention mechanisms. Therefore, it is reasonable to believe that the mechanism’s determination of feature weights is random and not really an enhancement of the important features.

#### 3.2.2. Experimental Part 2

We employ UMAP (uniform manifold approximation and projection) for feature visualization. UMAP, a novel manifold learning technique for dimensionality reduction, excels in visualization and scalability. It is grounded in Riemannian geometry and algebraic topology, offering enhanced preservation of global structure, superior runtime performance, and scalability [25]. Essentially, the effectiveness of the channel attention mechanism is judged by observing whether the similarity of the features matches the weights of the features. Typically, more similar features warrant similar weighting.

### 3.3. Experimental Results

#### 3.3.1. Vaihingen Dataset Results

Example 1 is shown in Figure 10. The scenario contains the full classification of the dataset. This figure primarily exhibits the processing of semantic information and the shading effects. In this scenario, all comparison methods, except the FEM and SEM, mistakenly classify a garden as a building. Additionally, in the lower-left corner, a tree obscuring a car goes undetected by most methods except the FEM and SEM. However, the SEM erroneously labels a piece of rubbish as a car, demonstrating an incorrect application of semantic information.

Example 2, shown in Figure 11, is designed to illustrate the different methods’ handling of details and semantic information. Additionally, this example shows how these methods deal with the effects of shadows. In this scene, accurately predicting the building at the bottom and the solitary tree on the left is challenging. The FEM has the best control of the details, accurately outlining the building and correctly identifying the adjacent bottom-left garden as an impermeable surface. It distinctly isolates the tree in question from others, avoiding misclassification or blending, unlike other methods. Although shadow effects are significant, the FEM still performs well. In comparison, the FEM more effectively predicts the buildings above the scene, reducing shadow effects and enhancing structural integrity. The scene also contains several cars, and as indicated by the red box in Figure 11, all methods except the FEM frequently misinterpret one car as two cars.

#### 3.3.2. OpenEarthMap Dataset Results

In example 3, shown in Figure 12, the main points of confusion are agricultural land versus grassland versus details of constructed land versus buildings. In this scenario, the FEM effectively discriminates between agricultural land and grassland, unlike other methods which show varying degrees of confusion. Regarding the buildings on the left side of the scene, the FEM provides the most accurate delineation of the buildings’ extent, whereas other methods incorrectly classify the developed land beneath the buildings as part of the buildings themselves.

Example 4, shown in Figure 13, emphasizes the interpretation of semantic information and shadow management. The scene is dominated by a prominent building in the center. A tree is located in the upper-right corner of this building, casting a shadow onto the pavement. Most methods, including the SEM and CBAM, misidentify the tree and its shadow as two separate trees. Others, like Unet and One-SEM, incorrectly categorize the tree as grass and its shadow as a tree. Only the FEM accurately identifies the tree’s position while correctly disregarding its shadow. In the scenario’s lower part, various methods mistakenly interpret agricultural land as water bodies, but the FEM significantly minimizes such errors.

### 3.4. Accuracy Metrics Results

From Table 1 and Table 2, it can be seen that the FEM has made good progress. In the Vaihingen dataset, the FEM achieves the best results in six indexes, including the mIoU, the OA, and the IoU of trees, as well as low vegetation, buildings, and impervious surfaces. Similarly, in the OpenEarthMap dataset, the FEM also achieves the best results in six indexes, including mIoU, the OA, and the IoU of trees, water, buildings, and agricultural land.

One-SEM achieved similar accuracy with the SEM, CBAM, and FCA, but One-FEM did not achieve similar accuracy with the FEM. From the visualization of the feature weights shown in Section 4.1, it can be seen that Unet with the addition of One-FEM is in fact equivalent to Unet. This equivalence is also substantiated by the similar performance of One-FEM and Unet in terms of accuracy, as shown in Table 1 and Table 2.

## 4. Discussion

Considering the different information contained in different bands, this paper designs a channel attention mechanism using feature information entropy. This newly proposed mechanism surpasses common channel attention mechanisms in both model accuracy and training efficiency. In the experiments on the Vaihingen dataset, the parameters of FEM only increase by 0.2 M, while the parameters of SEM, CBAM, and FCA increase by 0.9 M, 0.9 M, and 1.7 M, respectively. Furthermore, it can be a plug-and-play channel attention mechanism. Compared with recent studies, the FEM still secures similar accuracy gains. In the Vaihingen dataset, both the FEM and the VPA result in a 0.60% improvement in OA for the baseline network [19]. The FEM resulted in a 1.00% increase in mIoU for the baseline on the Vaihingen dataset, outperforming the results achieved by the MWM and CAM [18,26]; specifically, the MWM improved the mIoU of the baseline by 0.6%, and the CAM contributed a 0.2% increase to the mIoU of the baseline.

The proposed method can better promote the improvement of land use classification accuracy so that it can be better utilized in ecological monitoring, disaster prevention and mitigation, and urban planning. However, optimizing the design of the attention mechanism to guide the backbone network for superior feature extraction more effectively remains a primary focus. It is more common to combine the channel attention mechanism with the spatial attention mechanism [9,11], or different attention mechanisms can be designed for different tasks [27].

However, the current attention mechanism still has some shortcomings. The challenge of more effectively evaluating feature importance and directing the network’s focus towards these significant features remains an area for further exploration. Thus, this paper discusses the interpretability of the attention mechanism specifically through the visualization of the feature weights produced by the channel attention mechanism and the clustering of features.

### 4.1. Visualization of the Feature Weights

In the previous section, we described the design ideas for this part of the experiment. A tensor with all values being 1 was designed to determine if the channel attention mechanism is indiscriminate in enhancing features. The visualization of the feature weights obtained by the six attention mechanisms is shown in Figure 14 and Figure 15.

In both datasets, the weights obtained from the SEM, CBAM, and FCA using the multilayer perceptron architecture produce huge ups and downs in all four feature extraction stages. A similar situation also occurs in the weights obtained from One-SEM. Especially in the Vaihingen dataset, the weights tended to jump sideways between very large and very small values. These four channel attention mechanisms described above enhanced features in each stage of feature extraction, but in the end, some of the features were still judged as useless features. This situation illustrates the marginal inefficiency of the channel attention mechanisms for the multilayer perceptron architecture. Conversely, the FEM behaves differently. In the Vaihingen dataset, after the first stage of feature enhancement, all the features extracted afterwards are judged as important. This allows the channel attention mechanism not to be used in subsequent operations, saving computing resources and time and reducing model parameters. The number of parameters increases by only 0.2 M when using the FEM, while the SEM, CBAM, and FCA mechanisms increase the number of parameters by 0.9 M, 0.9 M, and 1.7 M, respectively. This performance is still observed in the OpenEarthMap dataset, where after four feature enhancements by the attention mechanisms, the SEM, CBAM, and FCA are still considered to have a large number of useless features.

By comparing One-FEM with One-SEM, we can find that the FEM does not respond to the tensor with all values being 1 to “enhance” the features. In contrast, the weights of the features can be obtained using One-SEM. These incorrect weights affect the features but do not significantly impact the accuracy, clearly demonstrating the inapplicability of the common attention mechanism in remote sensing.

### 4.2. Interpretability Analysis of Channel Attentional Mechanisms

To assess whether the channel attention mechanism correctly evaluates features, this paper carries out the following analyses. Normally, similar features should be judged equally important by the channel attention mechanism. Important features should be distinguishable from unimportant features. Then, we can check the correctness of the channel attention mechanism by observing the degree of similarity between features. However, the similarity of high dimensional features is difficult to measure and visualize. In order to visualize the similarity between features, this paper used Umap to compress the features into two-dimensional vectors, as shown in Figure 16 and Figure 17. Each point in these figures represents a downscaled feature. Measuring the Euclidean distance between different points can reflect the similarity of features to some extent. These features are classified based on the weights obtained by the channel attention mechanism and are divided into three groups: 0 to 0.333 (low), 0.333 to 0.667 (medium), and 0.667 to 1 (high).

It can clearly be noticed that both in the Vaihingen dataset and the OpenEarthMap dataset, the three classes of features are mixed haphazardly in cases where the SEM, CBAM, and FCA are used. In the Vaihingen dataset, the FEM categorizes the features more regularly, reflecting a strong separability. This indicates that the FEM correctly finds the weights of the features in contrast to the questionable weights assignment by the SEM, CBAM, and FCA. Moreover, a comparison of Figure 16 and Figure 17 reveals that in the Vaihingen dataset, the features are more densely clustered in specific regions, while in the OpenEarthMap dataset, feature scatter plots appear more dispersed. This may be related to more classes in the OpenEarthMap dataset.

To analyze the impact of the channel attention mechanism on feature enhancement more effectively, this study contrasts enhanced features with their unenhanced counterparts. The features are also visualized by using Umap, as shown in Figure 18. The SEM, CBAM, FCA and One-SEM demonstrate the similar situation: before feature enhancement, the three features of different importance are mixed together; while after feature enhancement, the three features of different importance become more easier to differentiate. The performance of the FEM proposed in this paper is more coherent, and the enhancement by the FEM results in the blurring of boundaries among the three important features This suggests that after FEM enhancement, the features of different importance become more similar to each other.

## 5. Conclusions

This paper introduces a novel channel attention mechanism termed the feature information entropy attention mechanism (FEM). The FEM utilizes the information content of features to assess their importance. Compared to the baseline Unet network, the mIoU improvement in the Vaihingen dataset is 1.00%, and in the OpenEarthMap dataset, it is 0.90%. Additionally, this study shows that the FEM outperforms other attention structures. When compared with the SEM, CBAM, and FCA, the mIoU improvements in the Vaihingen dataset are 0.90%, 1.10%, and 0.40%, respectively. In the OpenEarthMap dataset, these improvements are 2.30%, 2.20%, and 2.10%, respectively.

Through the visualization and analysis of features and their weights, the FEM exhibits advantages over common channel attention mechanisms. It can complete feature enhancement in the pre-feature extraction stage, and the different important features judged by the FEM can be well distinguished. This efficient and logical advantage allows the FEM to be better used in remote sensing.

## Figures and Tables

**Figure 1 sensors-24-01324-f001:**
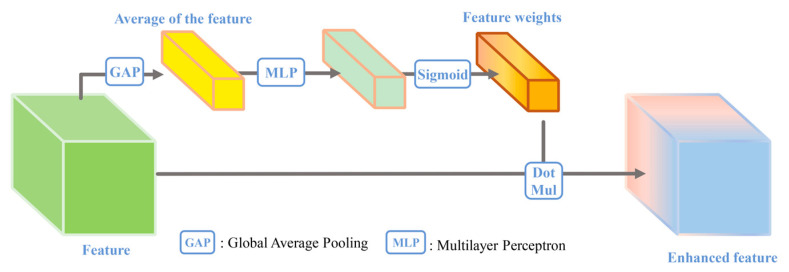
The squeeze-and-excitation module.

**Figure 2 sensors-24-01324-f002:**
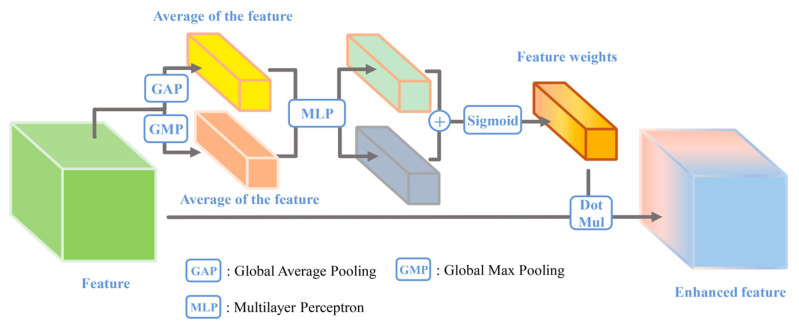
Convolutional block attention mechanism.

**Figure 3 sensors-24-01324-f003:**
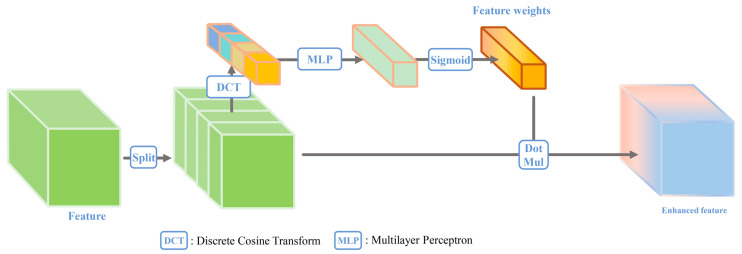
Frequency channel attention mechanism.

**Figure 4 sensors-24-01324-f004:**
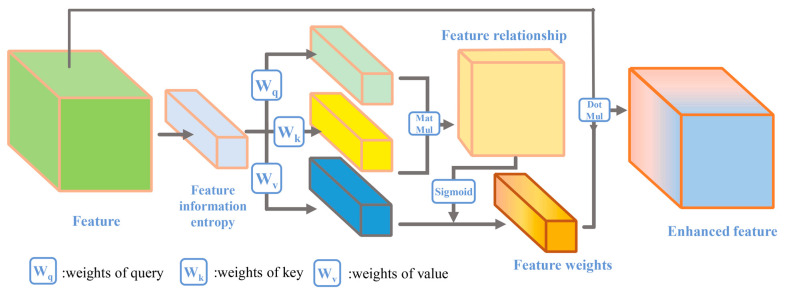
The feature information entropy attention module.

**Figure 5 sensors-24-01324-f005:**
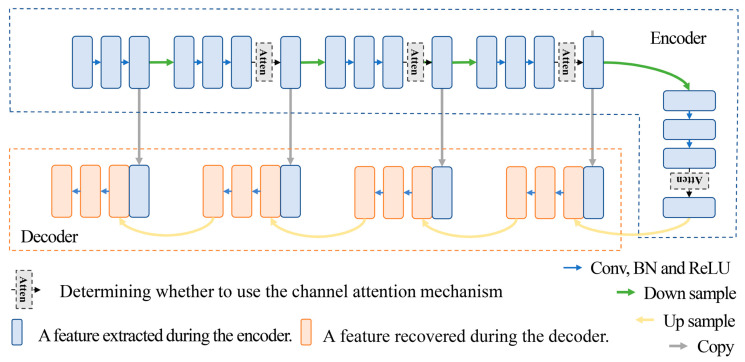
Unet network structure with various channel attention mechanisms.

**Figure 6 sensors-24-01324-f006:**
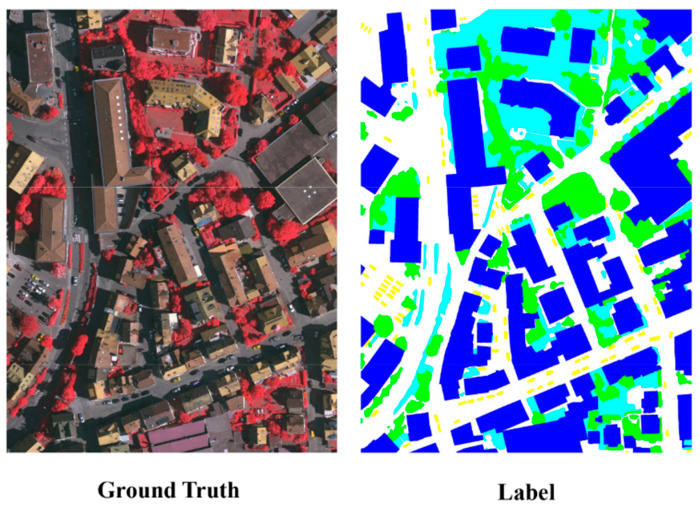
Example of Vaihingen dataset.

**Figure 7 sensors-24-01324-f007:**
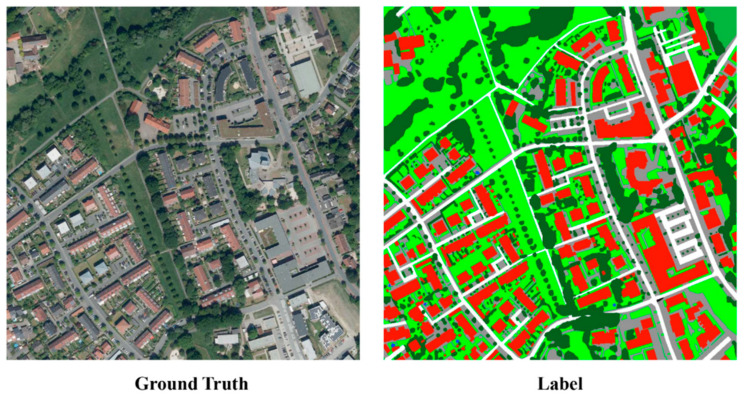
Example of OpenEarthMap dataset.

**Figure 8 sensors-24-01324-f008:**
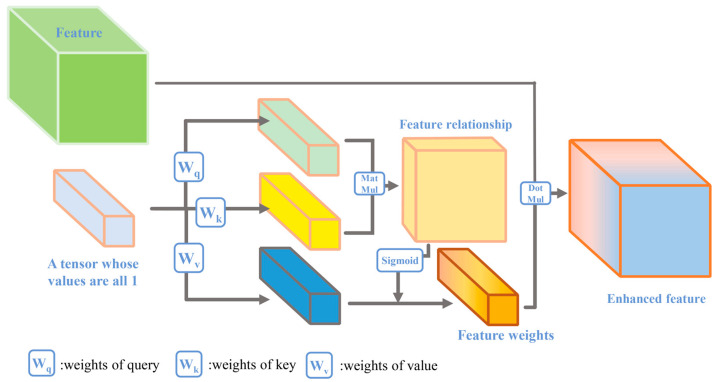
The feature entropy attention mechanism that replaced the feature statistics (One-FEM).

**Figure 9 sensors-24-01324-f009:**
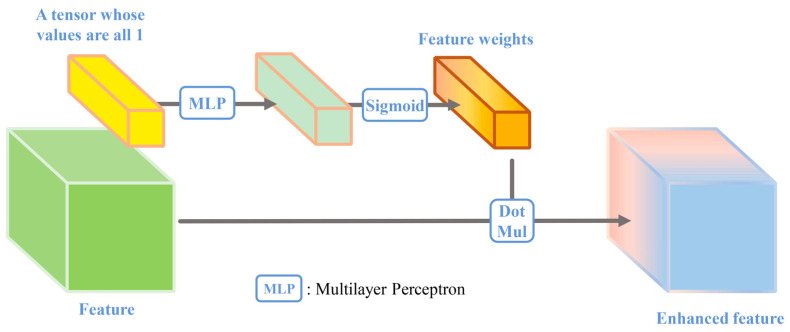
The squeeze excitation mechanism that replaced the feature statistics (One-FEM).

**Figure 10 sensors-24-01324-f010:**
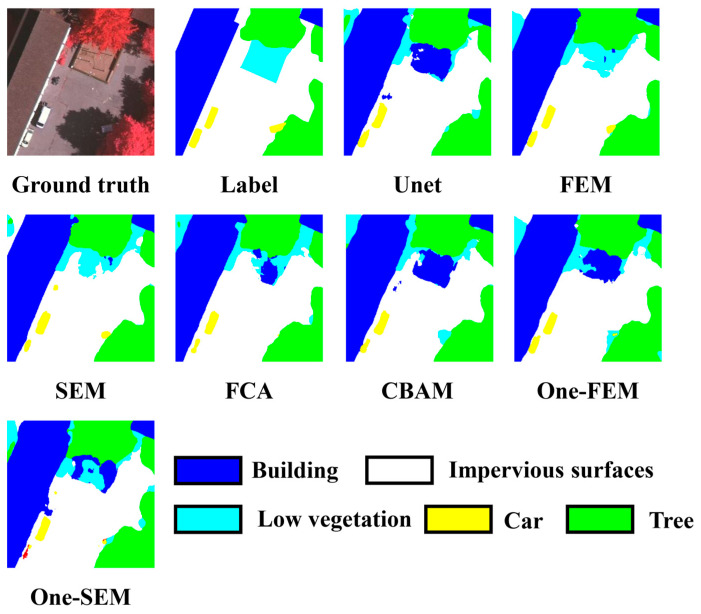
Comparing the performance of different channel attention mechanisms on the Vaihingen dataset (example 1).

**Figure 11 sensors-24-01324-f011:**
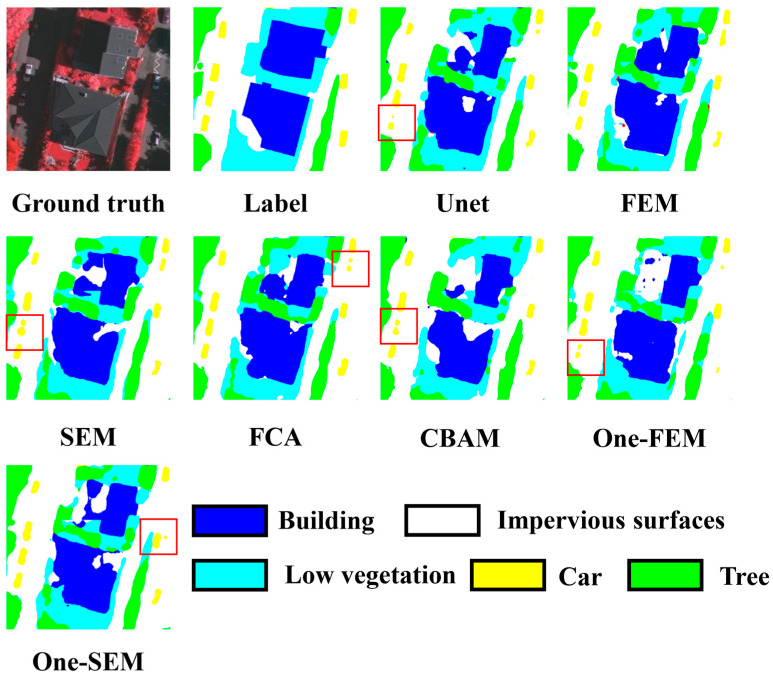
Comparing the performance of different channel attention mechanisms on the Vaihingen dataset (example 2).

**Figure 12 sensors-24-01324-f012:**
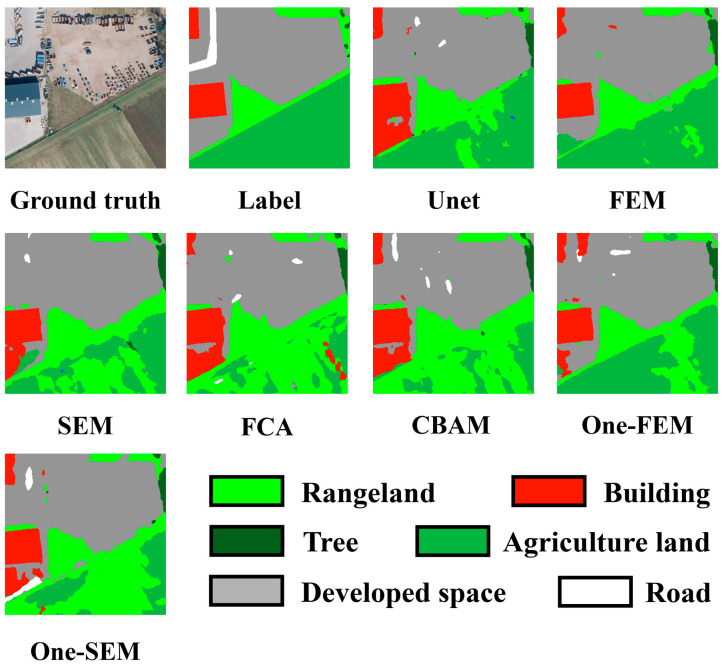
Comparing the performance of different channel attention mechanisms on the OpenEarthMap dataset (example 3).

**Figure 13 sensors-24-01324-f013:**
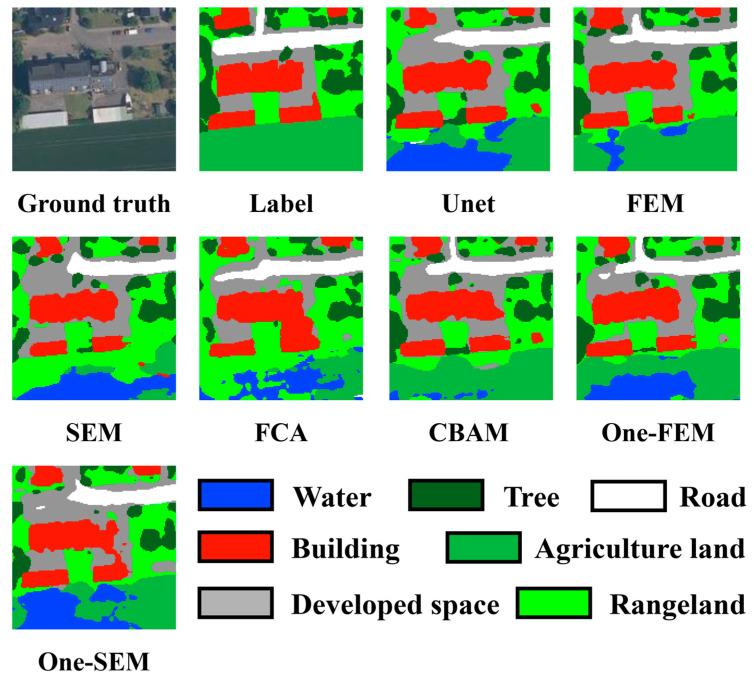
Comparing the performance of different channel attention mechanisms on the OpenEarthMap dataset (example 4).

**Figure 14 sensors-24-01324-f014:**
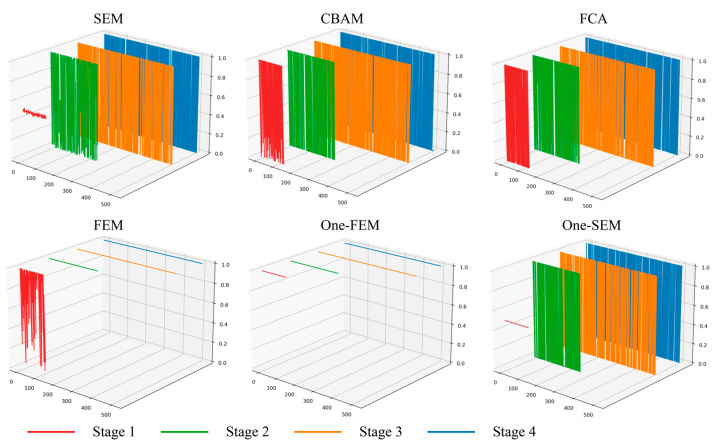
Demonstration of the weights produced by the six attention mechanisms in the Vaihingen dataset (data sourced from example 1 above).

**Figure 15 sensors-24-01324-f015:**
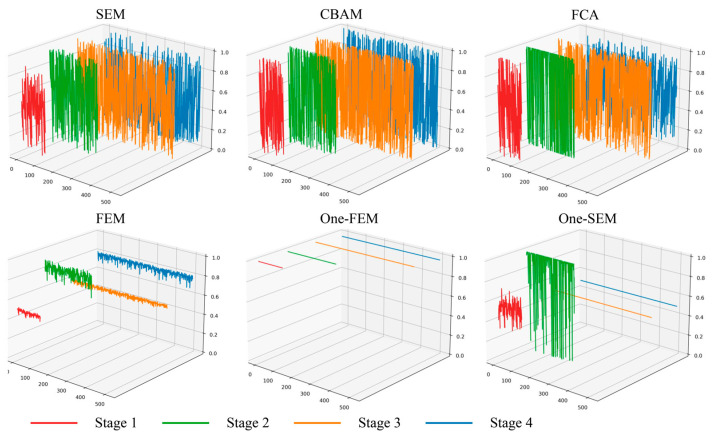
Display of the weights generated by the six attention mechanisms in the OpenEarthMap dataset (data sourced from example 3 above).

**Figure 16 sensors-24-01324-f016:**
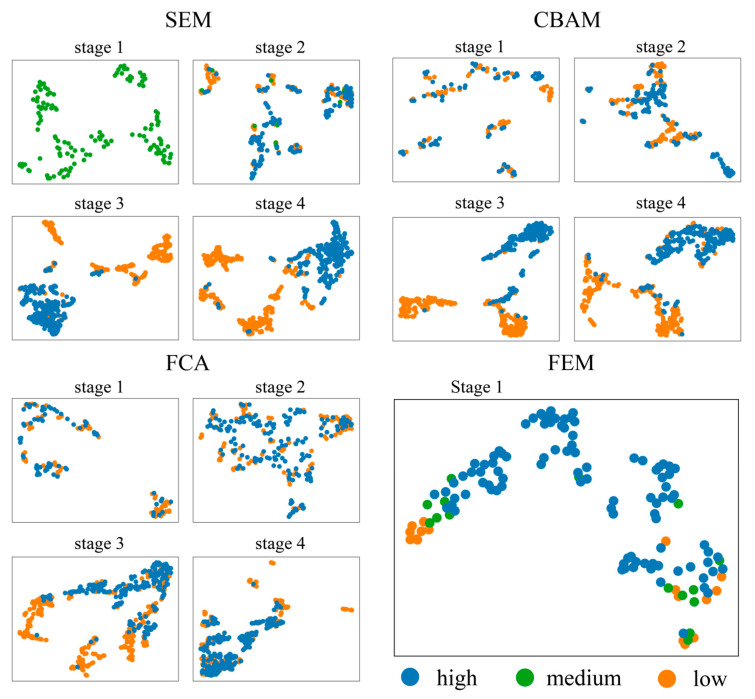
Visualization of different importance features after Umap dimensionality reduction (data sourced from example 1 above).

**Figure 17 sensors-24-01324-f017:**
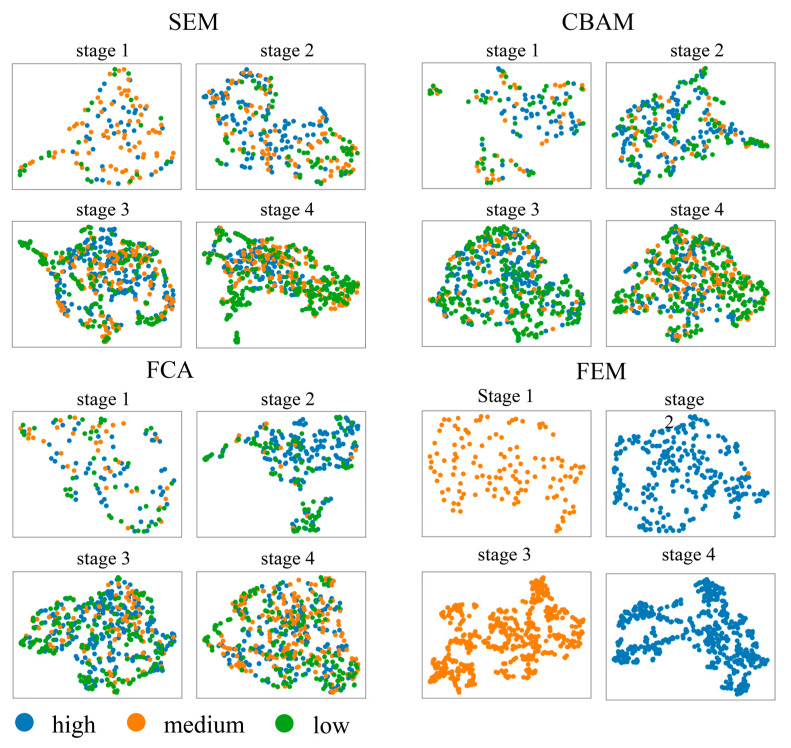
Visualization of different importance features after Umap dimensionality reduction (data sourced from example 3 above).

**Figure 18 sensors-24-01324-f018:**
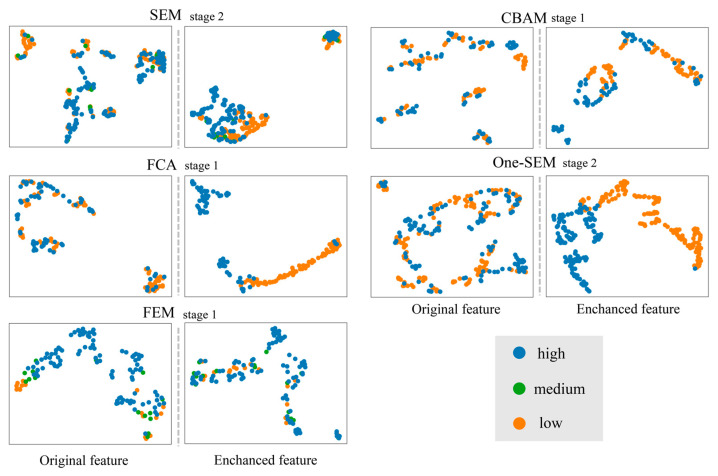
Comparison of original features with features enhanced by the channel attention mechanism (data sourced from example 1 above).

**Table 1 sensors-24-01324-t001:** Accuracy comparison of different channel attention mechanisms on the Vaihingen dataset.

	Accurate	Class IoU					mIoU	Overall Accuracy (OA)
Method		Tree	Car	Building	Low Vegetation	Impervious Surfaces		
FEM		**0.737**	0.571	**0.842**	**0.637**	**0.788**	**0.715**	**0.858**
SEM		0.732	0.557	0.837	0.622	0.783	0.706	0.853
CBAM		0.733	0.559	0.831	0.624	0.776	0.704	0.852
FCA		0.736	**0.573**	0.837	0.631	0.781	0.711	0.855
One-SEM		0.732	0.551	0.832	0.623	0.777	0.703	0.852
One-FEM		0.734	0.570	0.825	0.624	0.775	0.706	0.851
Unet		0.736	0.552	0.830	0.632	0.777	0.705	0.852

**Table 2 sensors-24-01324-t002:** Accuracy comparison of different channel attention mechanisms on the Vaihingen dataset.

	Accurate	Class IoU							mIoU	Overall Accuracy (OA)
Method		Building	Rangeland	Tree	Agriculture Land	Developed Space	Water	Road		
FEM		**0.783**	0.509	**0.685**	**0.75**	0.495	**0.270**	0.530	**0.574**	**0.763**
SEM		0.777	**0.528**	0.678	0.742	**0.515**	0.189	0.557	0.551	0.754
CBAM		0.782	0.51	0.681	0.724	0.505	0.221	0.550	0.552	0.750
FCA		0.776	0.509	0.65	0.704	0.486	0.199	0.551	0.553	0.754
One-SEM		0.782	0.498	0.672	0.71	0.495	0.23	**0.561**	0.564	0.758
One-FEM		0.78	0.499	0.68	0.686	0.5	0.207	0.554	0.558	0.756
Unet		0.78	0.504	0.67	0.726	0.491	0.23	0.554	0.565	0.759

## Data Availability

The code can be obtained from “https://github.com/bananaE5000/The-feature-information-entropy-attention-mechanism-FEM-”. The Vaihingen dataset can be obtained from “https://www.isprs.org/education/benchmarks/UrbanSemLab/2d-sem-label-vaihingen.aspx”. OpenEarthMap dataset can be obtained from “https://open-earth-map.org/”.

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
