# Peer review of "Semantic Segmentation of Remote Sensing Data Based on Channel Attention and Feature Information Entropy"

_sensors, 2024, doi:10.3390/s24041324_

Round 1

Reviewer 1 Report

Comments and Suggestions for Authors

It is suggested to move the details of the related studies from the Introduction to a related work section.

Introduction: emphasize the existing limitations and challenges considering the “latest” studies and describe the novel contribution of this work.

Proofread the paper and identify the facts that require references. For example, in section 2.1, (SEM) requires a reference.

It would be better to include a new section 2 for related studies. Describe and analyse the features considered, techniques used and the results obtained in those studies. For example, you can consider studies such as follows, in identifying the land cover data. 

https://doi.org/10.1016/B978-0-323-85214-2.00009-4

The authors have described a set of models utilized in this study, with their architectures. However, It is better to include a process flow diagram as well, that shows the process that transforms the input into the output.

The scientific contribution of the paper is good.

Also, in the discussion, include a sub-section for the comparison of the proposed studies with the existing similar systems.

Discuss possible real-world applications, that can be utilised the proposed solution ( as future extensions)

Comments on the Quality of English Language

Proofread the paper

Reviewer 2 Report

Comments and Suggestions for Authors

I have the following concerns.

1, Futures information mechanism has been used in machine learning for a long time. Clarify and justify what your difference is.

2. It is not clear why the authors give equation (1) for the continuous case, when discrete data are considered in the article.

3. It is not given on what size training, test and validation samples the practical results were obtained.

4. Tables 1 and 2 show the accuracy estimates based on the volume of data.

5. As the comparison shows, the average profit of your method is 0.05. Visually, we will not see this. It is necessary to show the profit by another parameter, for example, in time or the number of operations.

6.References must be significantly supplemented with articles from 2022-2024 to confirm the relevance of your research.

Comments on the Quality of English Language

Minor editing of English language required

Reviewer 3 Report

Comments and Suggestions for Authors

See attachment for comments.

Comments on the Quality of English Language

The author needs to correct a few grammatical errors.

Round 2

Reviewer 2 Report

Comments and Suggestions for Authors

I am almost satisfied with the answers to my comments except for the second one.

Comments on the Quality of English Language

 Minor editing of English language required

Reviewer 3 Report

Comments and Suggestions for Authors

All my concerns have been solved and I recommend publishing this work.